# Alterations in the Human Plasma Lipidome in Response to Tularemia Vaccination

**DOI:** 10.3390/vaccines8030414

**Published:** 2020-07-24

**Authors:** Kristal M. Maner-Smith, Johannes B. Goll, Manoj Khadka, Travis L. Jensen, Jennifer K. Colucci, Casey E. Gelber, Carolyn J. Albert, Steven E. Bosinger, Jacob D. Franke, Muktha Natrajan, Nadine Rouphael, Robert A. Johnson, Patrick Sanz, Evan J. Anderson, Daniel F. Hoft, Mark J. Mulligan, David A. Ford, Eric A. Ortlund

**Affiliations:** 1Department of Biochemistry, Emory School of Medicine, Emory University, Atlanta, GA 30322, USA; kmaners@emory.edu (K.M.M.-S.); mkhadka@emory.edu (M.K.); jennifer.k.colucci@emory.edu (J.K.C.); 2The Emmes Company, Rockville, MD 20850, USA; jgoll@emmes.com (J.B.G.); tjensen@emmes.com (T.L.J.); cgelber@emmes.com (C.E.G.); 3Department of Biochemistry and Molecular Biology, Saint Louis University School of Medicine, St. Louis, MO 63104, USA; albertc@slu.edu (C.J.A.); franke.jacobd@gmail.com (J.D.F.); 4Division of Microbiology and Immunology, Emory University, Atlanta, GA 30322, USA; steven.bosinger@emory.edu; 5Emory Vaccine Center, Emory University, Atlanta, GA 30322, USA; muktha.natrajan@emory.edu (M.N.); nroupha@emory.edu (N.R.); evanderson@emory.edu (E.J.A.); mark.mulligan@nyulangone.org (M.J.M.); 6Department of Pathology and Laboratory Medicine, Emory University School of Medicine, Decatur, GA 30030, USA; 7Division of Infectious Diseases, Department of Medicine, Emory University School of Medicine, Atlanta, GA 30322, USA; 8Biomedical Advanced Research and Development Authority, US Department of Health and Human Services, Washington, DC 20201, USA; robert.johnson@hhs.gov; 9Division of Microbiology and Infectious Diseases, National Institute of Allergy and Infectious Diseases, National Institutes of Health, Rockville, MD 20892, USA; patrick.sanz@nih.gov; 10Department of Pediatrics, Emory University School of Medicine and Children’s Healthcare of Atlanta, Atlanta, GA 30322, USA; 11Department of Internal Medicine, Saint Louis University School of Medicine, St. Louis, MO 63104, USA; daniel.hoft@health.slu.edu; 12Division of Infectious Diseases and Immunology, Department of Medicine, and New York University (NYU) Langone Vaccine Center, NYU School of Medicine, New York, NY 10016, USA

**Keywords:** lipidomics, oxylipins, tularemia, 5-hydroxyeicosatetraenoic (5HETE), inflammation, vaccine response

## Abstract

Tularemia is a highly infectious and contagious disease caused by the bacterium *Francisella tularensis*. To better understand human response to a live-attenuated tularemia vaccine and the biological pathways altered post-vaccination, healthy adults were vaccinated, and plasma was collected pre- and post-vaccination for longitudinal lipidomics studies. Using tandem mass spectrometry, we fully characterized individual lipid species within predominant lipid classes to identify changes in the plasma lipidome during the vaccine response. Separately, we targeted oxylipins, a subset of lipid mediators involved in inflammatory pathways. We identified 14 differentially abundant lipid species from eight lipid classes. These included 5-hydroxyeicosatetraenoic acid (5-HETE) which is indicative of lipoxygenase activity and, subsequently, inflammation. Results suggest that 5-HETE was metabolized to a dihydroxyeicosatrienoic acid (DHET) by day 7 post-vaccination, shedding light on the kinetics of the 5-HETE-mediated inflammatory response. In addition to 5-HETE and DHET, we observed pronounced changes in 34:1 phosphatidylinositol, anandamide, oleamide, ceramides, 16:1 cholesteryl ester, and other glycerophospholipids; several of these changes in abundance were correlated with serum cytokines and T cell activation. These data provide new insights into alterations in plasma lipidome post-tularemia vaccination, potentially identifying key mediators and pathways involved in vaccine response and efficacy.

## 1. Introduction

Vaccines initially trigger innate immune responses that result in development of an adaptive immune response with establishment of immunological memory [1]. This process is incompletely understood but is probably best detailed for the Yellow Fever vaccine [2]. This vaccine is highly effective and elicits a wide range of immune responses, resulting in greater than 30 years of protection from yellow fever. Previously published research has shown that the vaccine produces a biologic signature identified by transcriptomics, cytokine analyses and flow cytometry [3]. Such biologic signatures may correlate with immunogenicity to predict the efficacy of novel vaccines. Lipids are key stress-response and immune signaling molecules and changes in their circulating concentrations reflect a host of immune processes [4,5,6]. Thus, lipids play a critical role in inflammation. Accordingly, performing lipidomics analysis is a key element in the systems-wide multi-omic platform approaches leveraging data from transcriptomics, metabolomics, and proteomics to obtain a comprehensive picture of the vaccine response.

Tularemia, caused by the intracellular Gram-negative bacterium *Francisella tularensis*, is transmitted from infected ticks and mosquitos to humans. *F. tularensis* has a mortality rate approaching 30% if untreated and is one of the most highly infectious bacteria known; infection with as few as 10 organisms can cause severe disease [7,8,9]. Due to this high infectivity, *F. tularensis* could be weaponized and used as a biological weapon. A live-attenuated tularemia vaccine has been used by the US Army for decades and can protect against severe disease. However, this vaccine is associated with significant reactivity and has not been widely used outside of at-risk populations. In addition, the US supply of this live-attenuated vaccine has been depleted. A prospective randomized study between a recently generated new and old lot of a tularemia vaccine showed similar immunogenicity [8]. To evaluate the full complement of biological responses to this new lot of vaccine, we performed a multi-omics approach to discover changes in genome, proteome, metabolome, and lipidome [10,11]. In this report, we explore changes in the plasma lipidome of vaccine recipients on day 1, 2, 7, 14 post-vaccination as compared to lipidome pre-vaccination, with aims to determine lipid biomarkers of serological responses.

## 2. Materials and Methods

### 2.1. Study Design

For this study, a subset of plasma samples from a tularemia vaccine clinical trial was used [8] (ClinicalTrials.gov identifier NCT01150695). In the original trial, healthy subjects aged 18 to 45 years old were recruited and vaccinated with a single, undiluted dose of the *Francisella tularensis* live vaccine strain (LVS) lot 20 produced by DynPort Vaccine Company (DVC-LVS) or another *F. tularensis* vaccine (USAMRIID-LVS) via scarification on day 0. Demographic information for study participants is shown in Appendix A. The vaccines were administered in the ulnar aspect of the volar surface (palm side) of the forearm midway between the wrist and the elbow (8). The protocol and consent form were reviewed by the US Food and Drug Administration and approved and monitored by the clinical site institutional review boards. Here, we investigated plasma samples from 10 subjects taken pre-vaccination (day 0) and on days 1, 2, 7, 14 following DVC-LVS vaccination to assess lipidomics responses. Plasma samples were collected in tubes containing di-potassium ethylenediaminetetraacetic acid (K_2_EDTA).

### 2.2. Standards

Internal synthetic standards were obtained from Avanti Polar Lipids, Alabaster, AL, USA. These include di-20:0 phosphatidylcholine (PC) (x:y where x indicates number of carbons and y indicates number of double bonds in fatty acid constituents), di-14:0 phosphatidylethanolamine (PE), N-17:0 sphingomyelin (SM), di-14:0 phosphatidylserine (PS), 17:0 lysophosphatidylcholine (LPC), 14:0 lysophosphatidylethanolamine (LPE), di-20:0 diacylglycerol (DAG), tri-17:0 triacylglycerol (TAG), and 17:0 free fatty acid (FFA), 17:0 cholesteryl ester (CE) and N-17:0 ceramide (Cer). Each of these internal standards were added at concentrations that approximate plasma concentrations in humans. Concentrations are reported in Appendix A.

### 2.3. Lipid Extraction

Targeted lipidomics experiments were conducted at two different mass spectrometry facilities using aliquots from the same subject and timepoint as a pilot of a larger consortia study. To reduce potential systematic errors, both laboratories performed the same lipid extraction technique and used a common internal standard mixture prepared at Saint Louis University.

A volume of 100 µL of plasma was extracted using a modified Bligh and Dyer lipid extraction [12]. For this, 100 µL plasma was homogenized in 500 µL 2:1 *v*/*v* methanol:chloroform and vortexed to ensure homogeneity of sample. To this, 0.1 mM sodium chloride was added to aid in the partitioning of zwitterionic lipids to the organic phase. The organic phase was then recovered, dried under nitrogen gas, and the lipid weight recorded. Recovered lipids were then reconstituted in 1 mL of 1:1 *v*/*v* chloroform:methanol prior to injection into the mass spectrometer. To optimize PE ionization efficiency, an aliquot of extracted lipid was used to convert PE to PE-fMOC derivatives [13].

For oxylipin analysis, 200 µL plasma was extracted using C18 solid phase extraction (SPE) cassettes, (Restek, Bellefont, PA, USA) [14,15,16,17]. Briefly, C18 cassettes were conditioned using ethyl acetate, methanol, and water, in succession. The sample was then deposited on the matrix and rinsed with 3 column volumes of water and hexane, respectively. Oxylipins were then eluted with 3 column volumes of methyl formate. Recovered lipids were then dried and the lipid weight was recorded. The samples were then suspended in 500 µL 1:1 chloroform:methanol prior to injection on the mass spectrometer.

### 2.4. Mass Spectrometry

Targeted lipidomics was conducted using triple quadrupole mass spectrometers, SCIEX QTRAP5500 (Framingham, MA, USA) and a Thermo Quantum (Waltham, MA, USA) at EMO and SLU sites, respectively. Lipids were directly infused into the mass spectrometer at a flow rate of 5 µL/min. Instrumental parameters were optimized for each lipid class using a single, shared internal standard and were held constant during the course of the experiment. A table of instrumental parameters is shown in Appendix A.

To determine the distribution of lipids in extracted plasma samples, shotgun methodology was conducted whereby user-specified lipid classes were selectively targeted using characteristic scans. PC species were quantified in the negative ion mode by neutral loss scanning at 50 amu, corresponding to the loss of methylene chloride from the chlorinated choline headgroup [18,19,20,21]. Similarly, SM molecular species were quantified as chlorinated adducts in the negative ion mode using neutral loss scanning of 50 amu [22]. PE molecular species were quantified as their fMOC derivatives in the negative ion mode using neutral loss scanning of 222.2 amu [22]. Cholesteryl ester (CE) molecular species were quantified as sodiated adducts in the positive ion mode using neutral loss scanning of 368.5 amu as previously described [23]. Ceramide (Cer) molecular species were quantified in the negative ion mode using neutral loss scanning of 256.2 amu [24]. A table of the targeted lipid classes and the scans used isolate these lipids is shown in Table 1. Characteristic transitions that are representative of key oxidized lipids were monitored and are presented in Table 2 [14,15,16,17,25].

### 2.5. Lipid Quantification

Targeted lipids were quantified in fmol/µL by dividing a sample’s peak intensity for a certain lipid by the corresponding internal control/standard lipid peak intensity and by multiplying it with the corresponding internal control/standard of known concentration. Corrections were also made for type I and type II ^13^C isotope effects [13]. Additional corrections were made from response curves for CE molecular species [23]. For oxylipins, peak intensity ratios were multiplied by the corresponding external control/standard’s molecular weight to obtain fmol/µL. Standard lipidomics nomenclature was used throughout whereby acyl linkages are standard and ether linked lipids were denoted as: p = plasmalogen subclass and e = alkyl ether subclass. Aliphatic groups in lipid classes were also denoted as x:y, where x = number of carbons and y = number of double bonds (e.g., 20:4 (arachidonic acid) has 20 carbons and 4 double bonds).

### 2.6. Data Analysis

#### Missing Values and Baseline Calculations

Observations with 0 fmol/µL were set to not applicable (NA)/missing and these values were imputed using the k-nearest neighbor algorithm implemented in the impute R package (Version 1.44.0) [26]. Lipids with at least 40/50 (80%) non-missing observations were used as input for imputation and downstream analysis. The number of neighbors to be used as part of the imputation step was set to 8. Subject-specific log_2_ lipid fold changes from baseline were calculated for each subject and post-vaccination day (day 1, 2, 7, 14) by subtracting baseline (day 0) imputed log_2_ fmol/µL from each of the subject’s post-vaccination day imputed log_2_ fmol/µL. After consolidating the data across experiments, the dataset included 129 unique lipids—of which, 116 had at least 80% non-missing observations. The complete list of identified lipids is provided in Appendix A.

### 2.7. Statistics

Data were analyzed using the R statistical programming language (Version 3.2.5) (R Foundation for Statistical Computing, Vienna, Austria) and R Bioconductor [27,28]. Lipids that significantly differed in their response from baseline were identified by using a two-sided permutation paired t-test comparing baseline (day 0) to post-vaccination (day x) lipid fmol/µL (H0: u _(day−day0)_ = 0, H1: u _(dayx − day0)_ ≠ 0; on the log_2_ scale). We used a combination of permutation p-value cut off of <0.05 and an effect size cut off of ≥1.2-fold difference between mean pre- and post-vaccination responses to determine significantly differentially abundant (DA) lipids for each post-vaccination day. Confidence intervals of the median fold change response for the time trend plots were calculated using bootstrapping (1000 samples each).

Associations between DA lipid log_2_ fold changes and cytokine log_2_ fold changes or peak T-cell and tularemia-specific microagglutination responses were assessed using Spearman correlation. Peak T-cell responses (CD3 + CD4 + CD38 + HLA-DR + cells and CD3 + CD8 + CD38 + HLA-DR + cells) were based on the peak percentage of activated cells post-vaccination on days 7, 14, or 28 (11). Peak tularemia-specific microagglutination titer was represented as the log_2_ transformed peak titer observed on days 14 or 28 as collected in [8].

### 2.8. Transcriptomics Data

We used transcriptomics data from our parallel study that assessed transcriptome-wide gene expression following-vaccination of PBMCs obtained from the same subjects and timepoints as in this study using RNA-Seq [11] GEO.

## 3. Results

To assess lipidomic responses following vaccination with a *Francisella tularensis* live vaccine strain (DVS-LVC), we utilized plasma samples from 10 subjects taken pre-vaccination (day 0) and on days 1, 2, 7, 14 following vaccination. Demographics are summarized in Appendix A. Extracted lipids were analyzed by tandem mass spectrometry, using both traditional shotgun lipidomics as well as MRM-based methods. A series of characteristic scans were performed to isolate each abundant lipid class present in plasma (Table 1 and Table 2). Overall, 129 lipids were identified (Appendix A), —of which, 116 lipids, consisting of 14 oxylipins, 75 phospholipids, 14 cholesteryl esters, and 13 sphingolipids, had sufficient non-missing data to be analyzed (Appendix A). Fourteen of the 116 lipids were found to be DA (*p* < 0.05, fold change ≥ 1.2) on day 1, 2, 7, or 14 post-vaccination compared to pre-vaccination (day 0) and are presented in Table 3.

A breakdown of these lipids according to classification is shown in Figure 1. Phospholipids showed the largest proportion of DA (50.0%), followed by oxylipins (28.6%), cholesteryl esters (14.3%), and sphingolipids (7.7%).

To investigate changes over time for differentially abundant (DA) lipids, we visualized DA lipid responses from pre-vaccination using radar plots. Figure 2 is a radar plot showing the magnitude of change of the 19 differentially abundant lipids identified. Summed DHET Species showed the largest increase in molar concentration compared to pre-vaccination overall with a 2-fold increase over pre-vaccination on day 7 post-vaccination. By contrast, 5-HETE showed diametrically opposed changes on day 14 following tularemia vaccination.

To further investigate changes in DHET species in relation to 5-HETE, we plotted log_2_ fold change time trends and associated 95% confidence intervals over time (Figure 3). Figure 3A outlines the mechanism whereby 5-HETE is converted to DHET through the enzymatic action of the enzyme encoded by the *CYP4F3* gene, while Figure 3B,C shows inverse fold changes of both species showing a maximum decrease for 5-HETE and a maximum increase for DHET on day 7 post-vaccination and returning to close to baseline levels on day 14.

In addition to bioactive oxylipins, CE abundance was evaluated. Figure 4 is a time trend plot that probes the plasma abundance of DA lipid 16:1CE versus transcript of *ABCA1*, a protein target responsible for cholesterol export. The abundance of 16:1CE decreased on day 2 post-vaccination. In addition, the *ABCA1* gene encoding for the primary protein responsible for cholesterol efflux from macrophages and vascular tissue to circulating high-density lipoprotein showed a significant decrease at all post-vaccination days except day 2, demonstrating that cholesterol homeostasis mechanisms may also be affected by tularemia vaccination.

In total, 50% of the identified DA lipids were phospholipids, some containing esterified biologically active lipids such as arachidonic acid (20:4) and oleic acid (18:1) (Figure 5). These lipids are involved in inflammatory pathways leading to the conversion of 20:4 and 18:1 to endocannabinoids AEA and OEA, respectively, as outlined in Figure 5A. The plasma abundance of these lipids (Figure 5B,C) reveals that both OEA and AEA decrease on day 2 post-vaccination and returning to baseline by day 14.

To further characterize the role of DA lipids in the vaccine response, we generated correlation networks between log_2_ fold changes of 14 DA lipids, 22 cytokines, peak CD4/CD8 T cell activation, and tularemia-specific microagglutination using Spearman correlation (Figure 6). A list of measured serum cytokines and their abbreviations is in supplement (Appendix A).

On day 1, CD4+ T-cell activation was negatively correlated with change in molar concentration of 16:1 LPC which was decreased (Figure 6). On day 2, changes in molar concentration of OEA and AEA in plasma correlated with anti-inflammatory cytokines interleukin-10 (IL-10) andinterleukin-13 (IL-13) and CD8 lymphocytes. In contrast, changes in OEA and AEA were negatively correlated with later peak CD8+ T-cell activation (Figure 6). On day 7, changes in OEA and AEA were associated with pro-inflammatory cytokine IL-6. The decrease in 5-HETE was positively associated with IL-6 as well. By day 7, CD8 was negatively associated with 34:1 PI and any associations with OEA and AEA were no longer seen. On day 14, change in DHET abundance was negatively associated with anti-inflammatory cytokines IL-10 and IL-13 and positively correlated with granulocyte colony-stimulating factor (G CSF) (Figure 6). 16:1 LPC showed a negative correlation with CD8+ cell activation. No correlation between DA lipid changes and microagglutination titer was observed with Spearman correlation ≥0.6 at any post-vaccination day.

## 4. Discussion

Lipids are a large and diverse family of molecules with considerable structural and biological diversity that play integral roles in cell structure, metabolism and signaling. The abundance of several lipid classes is differentially affected by administration of the tularemia vaccine including structural lipids like cholesterol esters and glycerophospholipids and signaling lipids such as oxylipins and ceramides.

Cholesterol is the most abundant sterol in human plasma and exists mainly as cholesterol esters circulating in lipoprotein pools. Currently, nearly 22 cholesterol ester species have been identified in plasma [29]. Here, we observed significant changes in the abundance of 16:1 CE on days 1 and 2 post-vaccination (Figure 4B). In the absence of pharmacological intervention, loss of CE may indicate diminished hepatic cholesterol biosynthesis, which is one of the major contributors to plasma cholesterol levels carried by lipoproteins [30,31,32]. Indeed, transcriptomics data from a parallel study conducted using aliquots of plasma from the same patients, confirmed that the *ABCA1* gene encoding for the protein that exports excess cholesterol from cells was indeed significantly down-regulated on days 1, 7, and 14, with decreasing expression over time (43%, 48%, and 59% reduction, respectively, Figure 4B) [11]. The reduction in CE abundance may also be explained by toll-like receptor (TLR) signaling-mediated changes in immune cell cholesterol metabolism [33]. Previous reports have shown that activation of TLR signaling leads to decreased cholesterol efflux, resulting in accumulation of cholesterol and amplification of inflammatory response [34]. Our findings are consistent with this report. In a parallel study, where transcriptomics was conducted on a separate aliquot of plasma collected from these same patients, TLR signaling was increased on day following vaccination with increased expression for genes encoding for TLR1 (diacylated and triacylated lipopeptide recognition), TLR5 (flagellin pattern recognition), and TLR6 (microbe-associated molecular pattern recognition) receptor proteins [11].

In addition to CEs, the plasma concentrations of select oxylipins were also differentially abundant over the course of the experiment. These are DHET species as well as their biosynthetic precursor, a pro-inflammatory oxylipin, 5-HETE. The 5-hydroxyeicosatetraenoic acid (5-HETE) is an eicosanoid that acts as an autocrine and paracrine signaling agent that contributes to the up-regulation of acute inflammatory and allergic responses [25,35]. The 5-HETE and its stereoisomers stimulate cells through the binding and activation of the G-protein coupled receptor (GPCR) oxoeicosanoid receptor 1 (OXER1) [35]. In turn, OXER1 activates the mitogen activated protein kinase/extracellular-signal related kinase (MAPK/ERK) pathway, p38 mitogen-activated protein kinases, cytosolic phospholipase A2, phosphatidylinositol 3-kinase/ protein kinase B (PI3K/AKT), protein kinase C ß/Ɛ, and ionic calcium channels. The 5-HETE stimulates its target cells to degranulate in order to release anti-bacterial cytokines, produce bactericidal and tissue-injuring reactive oxygen species (ROS), and activate the functions of the innate immune system [36]. The 5-HETE has also been shown to activate Peroxisome Proliferator-Activated Receptor (PPAR) isoform gamma, which is expressed in macrophages and dendritic cells and plays an important role in resolving inflammation [37,38]. Transcriptomics responses, as measured using RNA-Seq, indicated increasing pathway enrichment for several published PPARγ-related Immunologic Signature Gene Sets including GSE25123 (gene expression signals in macrophage-specific PPARγ knockout mice) [39] with increasing enrichment signals over time. Signals increased from 15 to 17, 39, and 51 overlapping differentially expressed (DE) genes for days 1, 2, 7, and 14, respectively [11].

The lifetime of 5-HETE is regulated through enzymatic conversion to inactive or less active metabolites, dihydroxyeicosatetraenoic acids (DHETs or diHETEs) by the Cytochrome P450F family of proteins including *CYP4F3* (Figure 3). DHETs are derived from arachidonic acid (ARA) and are created from the hydrolysis, epoxygenation and, subsequent reduction of ARA from membrane lipids as a part of the cytochrome P450 epoxygenase pathway. The up-regulation of this lipid has been shown in patient samples who have high degrees of meta-inflammation [25]. DHETs have also been shown to contribute to coronary vasodilation in micromolar (non-physiological) concentrations [25].

Here, we report statistically significant changes in plasma abundance of 5-HETE and DHETs. By day 7 post-vaccination, we observed 61% decreased abundance of 5-HETE and increased abundance of DHET (Figure 3). We interpret the decrease in 5-HETE to be representative of increased metabolism of this species into the less active metabolite, DHET. This suggests the resolution of inflammatory response mediated by this class of molecules. These data are corroborated by transcriptomics data that demonstrated that gene expression for *CYP4F22* was up-regulated on day 2 post-vaccination (1.7-fold increase), day 7 post-vaccination (1.6-fold increase), and day 14 post-vaccination (1.8-fold increase) compared to pre-vaccination [11]. Together, these data suggest a clearing of acute inflammatory molecules via breakdown to their less active metabolites, and thus a termination of the innate immune response.

In addition to 5-HETE, two additional bioactive lipid mediators were identified as differentially abundant over the time course of the study. These are OEA and AEA (Figure 5), which are endogenous ligands of PPARα, a widely expressed transcription factor whose ligand-mediated activation inhibits increases in pro-inflammatory mediators such as tumor necrosis factor-alpha (TNF-a), interleukin 1 beta (IL-1b), interleukin 6 (IL-6), and others [40]. OEA, the most potent of these, is synthesized in enterocytes and begins with the N-acylation of oleic acid (18:1) from membrane PCs, such as those measured in this study (i.e., 34:1 PC and 36:1 PC) (Appendix A, TULIPID038 and TULIPID044, respectively). These PC species were quantified but not determined to be differentially abundant.

Like OEA, AEA is also synthesized from membrane lipids. Membrane phospholipid species that contain arachidonate at the sn-2 position are acted upon by several enzymes in distinct pathways to liberate N-acylated arachidonate, which is later converted to AEA [41]. AEA is a potent activator of the cannabinoid receptor as well as PPARα, and both receptors have well documented roles in inflammation, [17,41]. In the present study, the plasma concentrations of both OEA and AEA significantly increase on days 7 and 14 in comparison to day 0. Combined, this suggests a resolution of inflammation and is consistent with changes in the abundance of 5-HETE.

Overall, this analysis showed that vaccination results in significant changes in the lipidome and that peak changes are observed seven days post-vaccination. Lipids with the greatest changes play a role in mediating inflammation. Moreover, oxylipin data suggest a clearing of acute inflammatory molecules (5-HETE) via breakdown to their less active metabolites (DHET) and thus a termination of the innate immune response on day 7. Similarly, decreased levels of cholesteryl esters accompanied by down-regulation of *ABCA1* gene expression indicates pro-inflammatory pathways may be involved including mechanisms for cholesterol removal via the *ABCA1* transporter. This was corroborated by correlation with the abundance of key pro-inflammatory cytokines as shown in Figure 6. There was an inverse correlation between abundance of the pro-inflammatory TNF-α cytokine and AEA on day 2. Similarly, on day 2, there is a negative correlation between AEA and CD8+ T cells, implying heightening of immune response post-vaccination [2,42]. Similarly, an increase in the abundance of OEA occurs with a decrease in the pro-inflammatory IL-6 cytokine on day 7, in line with prior studies and suggesting the resolution of inflammation [43,44]. While no associations were identified between DA lipid responses and tularemia-specific microagglutination titer, CD4+ T-cell activation was linked to DA lipids on day 1 and CD8+ T-cell activation was associated with DA lipid changes on days 2, 7, and 14. The fact that no association between lipids and microagglutination was observed indicated that DA lipids were not likely linked with peak antibody responses. However, the multiple associations with peak CD8+ cell activation imply that changes in plasma lipids might be directly or indirectly associated with activation of T-cells known to respond to intracellular pathogens including *F. tularensis* [45].

## 5. Conclusions

Taken together, we report that cholesterol esters, oxylipins, and certain glycerophospholipid species were differentially abundant post-tularemia vaccination, especially during the innate immune phases of vaccine response, suggesting that these lipids may be useful as future biomarkers of serological response to vaccination.

## Figures and Tables

**Figure 1 vaccines-08-00414-f001:**
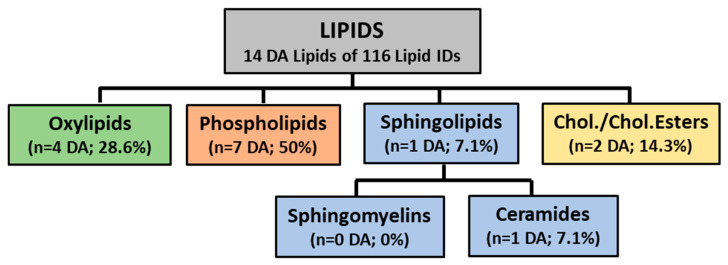
Schematic of lipid classes identified by targeted lipidomics. The following breakdown shows the number of differentially abundant lipids within each quantified lipid classes. Numbers in parentheses show the number of differentially abundant lipids identified within that class and percentages reflect the contribution of those lipids to total number of differentially abundant lipids. Phospholipids contained the largest proportion of differentially abundant lipids (seven DA lipids, 50%), followed by oxidized lipids (four DA lipids, 28.6%), cholesterol esters (two DA lipids, 14.3%), and sphingolipids (one DA lipid, 7.1%).

**Figure 2 vaccines-08-00414-f002:**
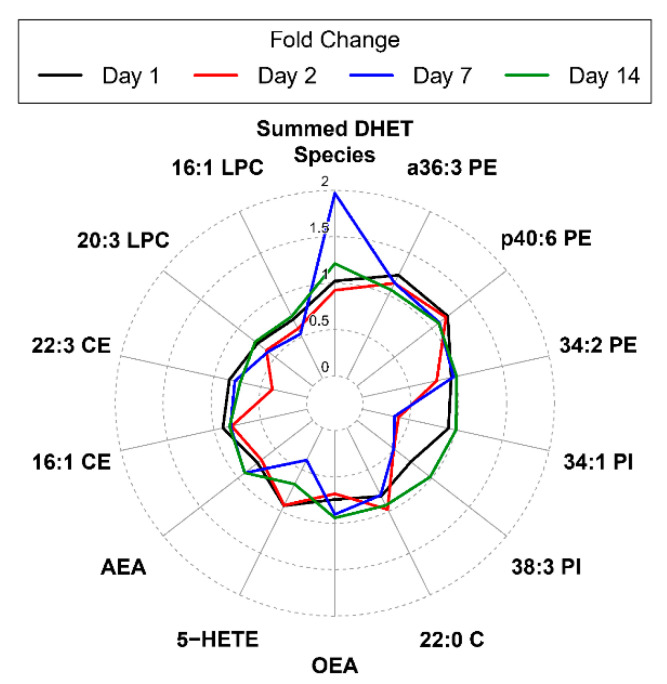
Radar plot summarizing mean fold change from pre-vaccination by day and DA lipids. Lines represent the magnitude of the mean fold change of differentially abundant lipid species on day 1, 2, 7, and 14 compared to pre-vaccination. Lipids are ordered clockwise by descending maximum absolute log_2_ fold change starting at the top center. The Summed DHET Species represents fold changes for the aggregated results of all DHET species lipids at each post-vaccination day vs. pre-vaccination.

**Figure 3 vaccines-08-00414-f003:**
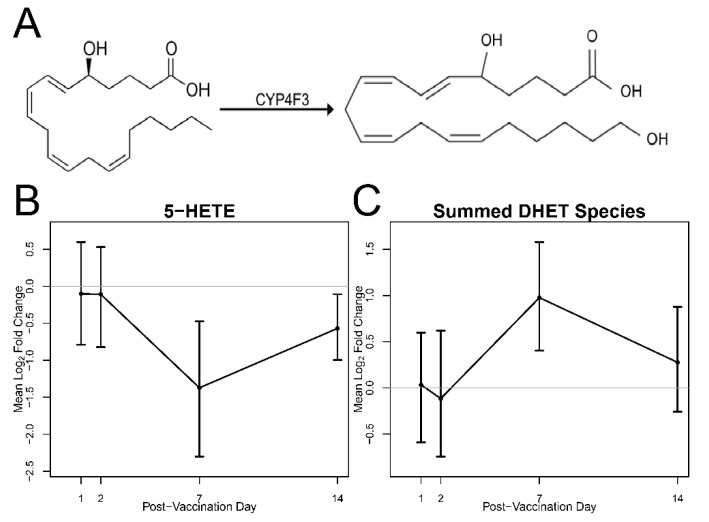
The 5-HETE is metabolized by CYP4F3 to form 5,20-DHET. (**A**) 5-HETE is converted to DHET species by CYP4F3. (**B**,**C**) 5-HETE and DHET Time trends of mean fold change from pre-vaccination and associated 95% bootstrap CIs. The abundance of 5-HETE and combined DHET species is found to be differentially abundant at 7 and 14 days post-vaccination.

**Figure 4 vaccines-08-00414-f004:**
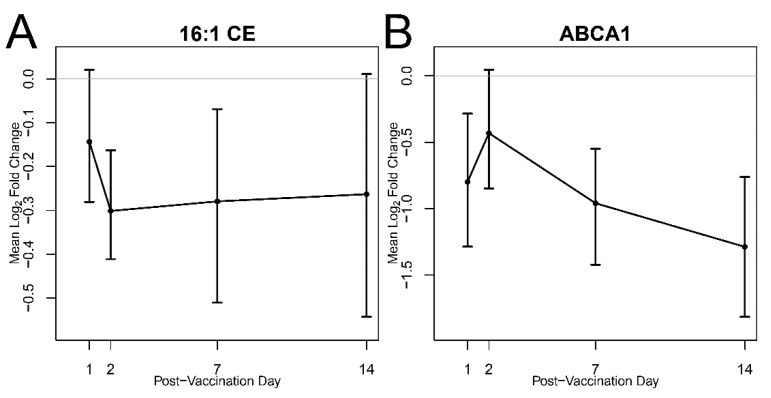
Reduction in cholesterol esters compared to pre-vaccination. (**A**) Time trend of mean log_2_ fold change of molar concentration of 16:1 CE on day 1, 2, 7, and 14 compared to pre-vaccination and associated 95% bootstrap CIs. (**B**) Time trend of *ABCA1* gene expression on day 1, 2, 7, and 14 relative to pre-vaccination. Gene expression of the *ABCA1* gene was significantly decreased at all post-vaccination days but day 2.

**Figure 5 vaccines-08-00414-f005:**
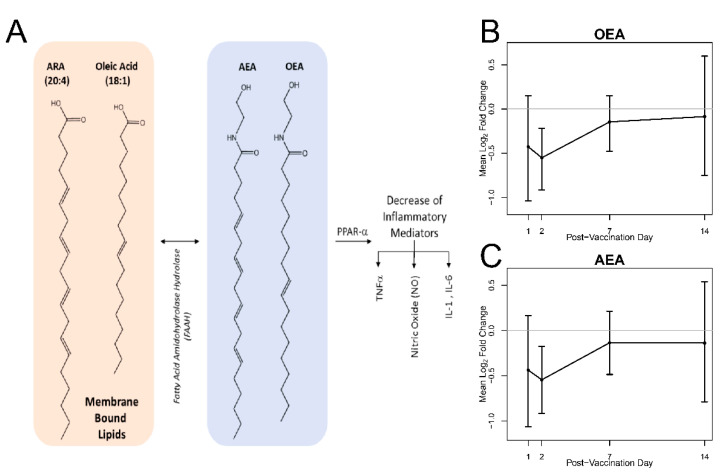
Biosynthesis of AEA and OEA from membrane lipids and plasma abundance. (**A**) AEA and OEA are synthesized from membrane bound lipids and are activators of PPAR-a, whose activation decreases cellular abundance of inflammatory mediators. Time trends of mean log2 fold change of (**B**) OEA and (**C**) AEA on day 1, 2, 7, and 14 relative to pre-vaccination and associated 95% bootstrap CIs. The plasma molar concentration of these species was found to be differentially abundant on day 2 when compared to pre-vaccination.

**Figure 6 vaccines-08-00414-f006:**
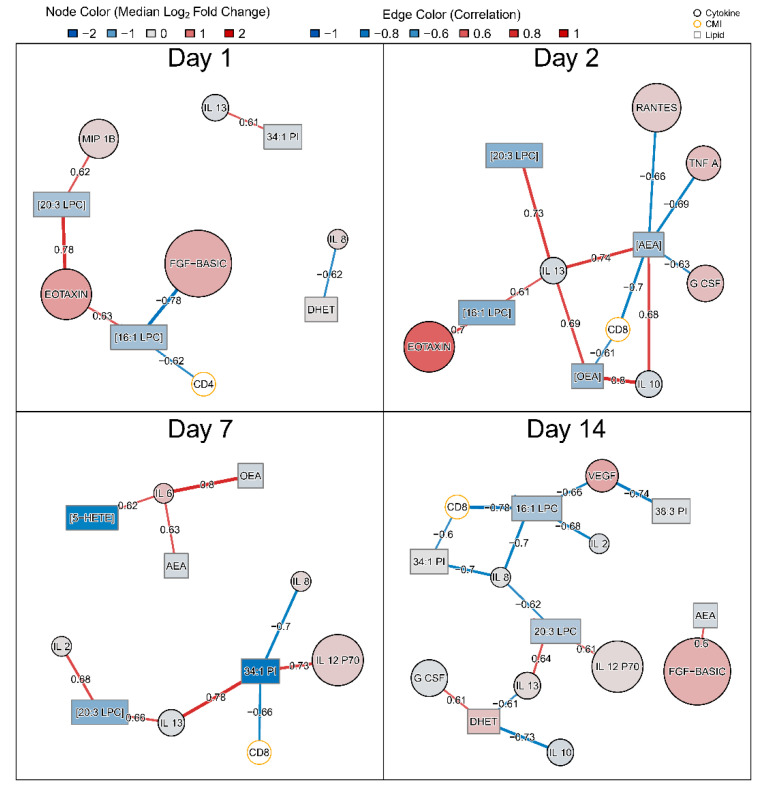
Spearman correlation network summarizing associations between DA lipid fold changes and changes in serum cytokines as well as peak T-cell activation. Pairwise Spearman correlations were assessed between fold changes of 14 DA lipids and 22 serum cytokines by post-vaccination day. In addition, correlations with peak CD4+ and CD8+ T-cell activation on days 7, 14, or 28 and tularemia-specific microagglutination were assessed. Black nodes represent lipids (rectangles) and cytokines (circles) while edges represent the Spearman correlation between fold changes. T-cell activation variables are shown in orange circles. Cytokine and lipid nodes are color-coded by log_2_ fold change. Edges are color coded and edge widths are scaled by Spearman correlation. To facilitate visual interpretation, networks were filtered for Spearman correlation ≥ 0.6.

**Table 1 vaccines-08-00414-t001:** Table of characteristic precursor scans used to selectively target select lipid classes present in plasma samples.

Lipid Class	Class Abbrev.	Ionization Mode	Characteristic Scan	Lipid IDs	Collision Energy
Phosphatidylethanolamine	PE	Negative	NL222	35	−30
Phosphatidylcholine	PC	Negative	NL50	42	−30
Sphingomyelin	SM	Negative	NL50	4	−24
Phosphatidylinositol	PI	Negative	Prec241	9	−30
Phosphatidylserine	PS	Negative	NL87	1	−30
Cholesteryl Esters	CE	Positive	NL368	14	25
Ceramide	Cer	Negative	NL256	11	−32

**Table 2 vaccines-08-00414-t002:** Oxidized Lipids Panel Transition Table. A multiple reaction monitoring (MRM)-based method was used to quantify the abundance of 14 oxidized lipid species. Species with the same transition are grouped and reported together.

Lipid Species	Abbreviation	m/z	Transition
Oleoylethanolamide	OEA	325.5	325 < 62
Arachidonoylethanolamine	AEA	347.5	347 < 62
Prostaglandin E2 Ethanolamide	PGE2 Ethanolamide	395.5	378 < 62
Prostaglandin F2a Ethanolamide	PGF2a Ethanolamide	397.5	380 < 62
12(13)-EpOME/13-HODE			
12(13)epoxy-9-octadecenoic acid	12(13)-EpOME	296.5	296 < 168
13-hydroxy-9,11-octadecadienoic acid	13-HODE	296.5	296 < 168
9,10-dihydroxy-12-octadecenoic acid	9,10-DiHOME	314.5	314 < 172
Thromboxane B2	TXB2	370.5	370 < 147
12-hydroxy-5,8,10,14-eicosatetraenoic acid	12-HETE	320.5	320 < 87
9-HETE/11(12)-EET			
9-hydroxy-5,7,11,14-eicosatetraenoic acid	9-HETE	320.5	320 < 167
11(12)-epoxy-5,8,14-eicosatrienoic acid	11(12)-EET	320.5	320 < 167
20-hydroxy-5,8,11,14-eicosatetraenoic acid	20-HETE	320.5	320 < 289
5-hydroxy-6,8,11,14-eicosatetraenoic acid	5-HETE	320.5	320 < 301
8(9)-epoxy-5,11,14-eicosatrienoic acid	8(9)-EET	320.5	320 < 69
14(15)-epoxy-5,8,11-eicosatrienoic acid	14(15)-EET	320.5	320 < 220
Summed DHET Species			
14,15-dihydroxy-5,8,11-eicosatrienoic acid	14,15-DHET	338.5	338 < 256, 170,184,82
11,12-dihydroxy-5,8,14-eicosatrienoic acid	11,12-DHET	338.5	338 < 256, 170,184,82
8,9-dihydroxy-5,11,14-eicosatrienoic acid	8,9-DHET	338.5	338 < 256, 170,184,82
5,6-dihydroxy-8,11,14-eicosatrienoic acid	5,6-DHET	338.5	338 < 256, 170,184,82

**Table 3 vaccines-08-00414-t003:** Overview of differentially abundant lipids by day. A two-sided permutation paired t-test was used to determine lipids with post-vaccination plasma molar concentrations that significantly differed compared to pre-vaccination. Lipids with a *p*-value < 0.05 and an increase/decrease from pre-vaccination of at least 20% (±1.2-fold) were deemed differentially abundant.

Lipid Name	Lipid ID	Lipid Class	Day	Fold Change	t-Statistic	*p*-Value
16:1 LPC	TULIPID017	PC	Day 1	0.709	−2.7	0.0137
20:3 LPC	TULIPID024	PC	Day 1	0.74	−5.5	0.002
36:3 PE	TULIPID103	PE	Day 1	1.238	1.8	0.0469
p40:6 PE	TULIPID119	PE	Day 1	1.214	2.4	0.0449
16:1 CE	TULIPID078	CE	Day 2	0.811	−4.7	0.0039
16:1 LPC	TULIPID017	PC	Day 2	0.597	−2.9	0.0234
20:3 LPC	TULIPID024	PC	Day 2	0.62	−3.8	0.0078
22:3 CE	TULIPID089	CE	Day 2	0.376	−2.6	0.0371
34:1 PI	TULIPID058	PI	Day 2	0.388	−2.7	0.0254
34:2 PE	TULIPID099	PE	Day 2	0.797	−3.4	0.0098
AEA	TULIPID002	OXY	Day 2	0.686	−2.7	0.0312
OEA	TULIPID001	OXY	Day 2	0.682	−3.0	0.0156
16:1 CE	TULIPID078	CE	Day 7	0.824	−2.2	0.0371
16:1 LPC	TULIPID017	PC	Day 7	0.534	−2.6	0.0312
20:3 LPC	TULIPID024	PC	Day 7	0.598	−3.0	0.0195
22:0 Cer	TULIPID072	Cer	Day 7	0.807	−2.9	0.0254
38:3 PI	TULIPID064	PI	Day 7	0.491	−2.4	0.0469
5-HETE	TULIPID012	OXY	Day 7	0.386	−2.7	0.0312
Summed DHET Species	TULIPID007	OXY	Day 7	1.969	3.1	0.0215
5-HETE	TULIPID012	OXY	Day 14	0.674	−2.3	0.0488

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
