# Peer review of "Alterations in the Human Plasma Lipidome in Response to Tularemia Vaccination"

_vaccines, 2020, doi:10.3390/vaccines8030414_

Round 1

Reviewer 1 Report

In the manuscript by Maner-Smith et al., the authors investigated alternations of lipids in human plasma from subjects underwent Tularemia vaccination, and found that the changes in abundance of certain lipids were associated with innate or adaptive immune responses induced by vaccination. Their findings suggest the lipids as the potential biomarkers for monitoring the immune responses to Tularemia vaccination. The manuscript is well-written, and the results are comprehensively discussed. A few minor comments.

  1. The study subjects were a subset from a clinical trial. Though there were only 10 subjects, a detailed description on subject characteristic and clinical information should be included if they are available. 
  2. Which anticoagulant was used to obtain plasma?
  3. The Figure 6 may be more readable if a table having r and p for each correlation is provided as supplement. 

Reviewer 2 Report

Manuscript is well written and well presented in most parts. The introduction gives an appropriate background to the topic and sets out clearly the aims of the study. However, the paper is written in a way that does not present a self-contained piece of work that can be read and understood in isolation and without reference to other papers, which are part of this “multiomics” study and presumably submitted separately.

For this reason, I am unwilling to recommend the publication of this work in its current format.

As a minimum I would recommend the following changes:

  • An ethics statement about the clinical trial samples is not included. It is not appropriate to cite another publication that describes the trial data.
  • Figure 1 includes asterisks which can be misinterpreted as denoting statistical significance. The authors should consider using a different symbol
  • Figure 2 should be edited to make the diagram clearer e.g. summed DHET species labelled is too close to the legend and is a little confusing
  • Figure 4B summarises ABCA1 gene expression but no methods for these experiments are included. Moe importantly, there is no indication of the tissue source for these gene expression analyses, Gene symbols should be checked for correct formatting throughout.
  • P-values should be included for all significant changes in lipid or gene levels.
  • Rephrase sentence in line 252.
  • Line 268 – this list of serum markers is not appropriate. It should be included in a table and the materials and methods section.
  • There is no method included for the data presented in Figure 6. This figure needs to be significantly edited. It is illegible in its current presentation. It is not clear to me if this is the best way of presenting these data.
  • Line 306 is an inappropriate was of discussing reference 32. It should be changed to a traditional, more complete discussion of the work.
  • Line 309 – refers to data not presented in this paper.
  • The discussion needs significant re-writing after amendment of the rest of the paper. I am unable to follow easily the discussion in its current form.
